# Optimal integration of visual speed across different spatiotemporal frequency channels

**Matjaž Jogan and Alan A. Stocker**
Department of Psychology
University of Pennsylvania
Philadelphia, PA 19104
{mjogan,astocker}@sas.upenn.edu

## Abstract

How do humans perceive the speed of a coherent motion stimulus that contains motion energy in multiple spatiotemporal frequency bands? Here we tested the idea that perceived speed is the result of an integration process that optimally combines speed information across independent spatiotemporal frequency channels. We formalized this hypothesis with a Bayesian observer model that combines the likelihood functions provided by the individual channel responses (cues). We experimentally validated the model with a 2AFC speed discrimination experiment that measured subjects' perceived speed of drifting sinusoidal gratings with different contrasts and spatial frequencies, and of various combinations of these single gratings. We found that the perceived speeds of the combined stimuli are independent of the relative phase of the underlying grating components. The results also show that the discrimination thresholds are smaller for the combined stimuli than for the individual grating components, supporting the cue combination hypothesis. The proposed Bayesian model fits the data well, accounting for the full psychometric functions of both simple and combined stimuli. Fits are improved if we assume that the channel responses are subject to divisive normalization. Our results provide an important step toward a more complete model of visual motion perception that can predict perceived speeds for coherent motion stimuli of arbitrary spatial structure.

## 1   Introduction

Low contrast stimuli are perceived to move slower than high contrast ones [17]. This effect can be explained with a Bayesian observer model that assumes a prior distribution with a peak at slow speeds [18, 8, 15]. This assumption has been verified by reconstructing subjects' individual prior distributions from psychophysical data [16]. Based on a noisy sensory measurement $m$ of the true stimulus speed $s$ the Bayesian observer model computes the posterior probability

$$p(s|m) = \frac{p(m|s)p(s)}{p(m)} \qquad (1)$$

by multiplying the likelihood function $p(m|s)$ with the probability $p(s)$ representing the observer's prior expectation. If the measurement is unreliable (*e.g.* if stimulus contrast is low), the likelihood function is broad and the posterior probability distribution is shifted toward the peak of the prior, resulting in a perceived speed that is biased toward slow speeds. While this model is able to account for changes in perceived speed as a function of different internal noise levels (modulated by stimulus contrast), it does not possess the power to predict the influence of other factors known to modulate perceived speed such as for example the spatial frequency of the stimulus [14, 10, 2].

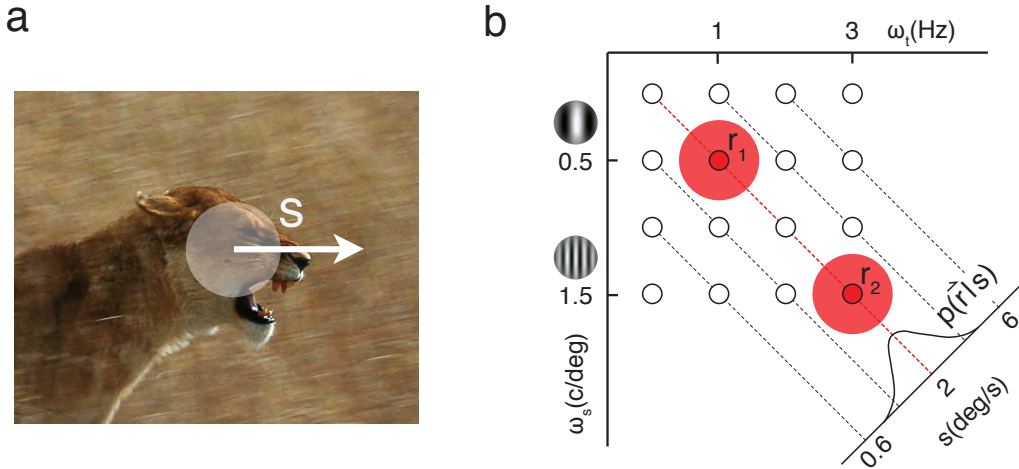

Figure 1: a) A natural stimulus in motion exhibits a rich spatiotemporal frequency spectrum that determines how humans perceive its speed $s$. b) Spatiotemporal energy diagram for motion in a given direction (*i.e.* speed) showing individual spatiotemporal frequency channels (white circles). A stimulus that contains spatial frequencies of $0.5\,\mathrm{c/deg}$ and $1.5\,\mathrm{c/deg}$ and moves with a speed of $2\,\mathrm{deg/s}$ will trigger responses $\vec{r} = \{r_1, r_2\}$ in two corresponding channels (red circles). The uncertainty about $s$ given the response vector $\vec{r}$ is expressed in the joint likelihood function $p(\vec{r}|s)$.

In this paper we make a step toward a more general observer model of visual speed perception that, in the longterm, will allow us to predict perceived speed for arbitrary complex stimuli (Fig. 1a). Inspired by physiological and psychophysical evidence we present an extension of the standard Bayesian model (Eq. 1), which decomposes complex motion stimuli into simpler components processed in separate spatiotemporal frequency channels. Based on the motion energy model [1, 12], we assume that each channel is sensitive to a narrow spatiotemporal frequency band. The observed speed of a stimulus is then a result of combining the sensory evidence provided by these individual channels with a prior expectation for slow speeds. Optimal integration of different sources of sensory evidence has been well documented in cue-combination experiments using cues of different modalities (see *e.g.* [4, 7]). Here we employ an analogous approach by treating the responses of individual spatiotemporal frequency channels as independent cues about a stimulus' motion.

We validated the model against the data of a series of psychophysical experiments in which we measured how humans' speed percept of coherent motion depends on the stimulus energy in different spatial frequency bands. Stimuli consisted of drifting sinusoidal gratings at two different spatial frequencies and contrasts, and various combinations of these single gratings. For a given stimulus speed $s$, single gratings target only one channel while the combined stimuli target multiple channels. A joint fit to the psychometric functions of all conditions demonstrates that our new model well captures human behavior both in terms of perceptual biases and discrimination thresholds.

## 2 Bayesian model

To define the new model, we start with the stimulus. We consider $s$ to be the speed of locally coherent and translational stimulus motion (Fig. 1a). This motion can be represented by its power spectrum in spatiotemporal frequency space. For a given motion direction the energy lies in a two-dimensional plane spanned by a temporal frequency axis $\omega_t$ and a spatial frequency axis $\omega_s$ and is constrained to coordinates that satisfy $s = \omega_t/\omega_s$ (Fig. 1b; red dashed line). According to the motion energy model, we assume that the visual system contains motion units that are tuned to specific locations in this plane [1, 12]. A coherent motion stimulus with speed $s$ and multiple spatial frequencies $\omega_s$ will therefore drive only those units whose tuning curves are centered at coordinates $(\omega_s, \omega_s s)$.

We formulate our Bayesian observer model in terms of $k$ spatiotemporal frequency channels, each tuned to a narrow spatiotemporal frequency band (Fig. 1b). A moving stimulus will elicit a total response $\vec{r} = [r_1, r_2, ..., r_k]$ from these channels. The response of each channel provides a likelihood

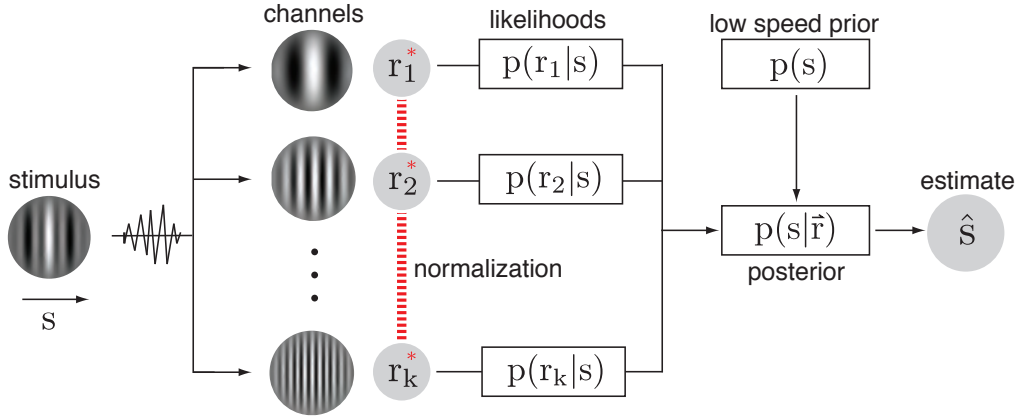

Figure 2: Bayesian observer model of speed perception with multiple spatiotemporal channels. A moving stimulus with speed $s$ is decomposed and processed in separate channels that are sensitive to energy in specific spatiotemporal frequency bands. Based on the channel response $r_i$ we formulate a likelihood function $p(r_i|s)$ for each channel. The posterior distribution $p(s|\vec{r})$ is defined by the combination of the likelihoods with a prior distribution $p(s)$. Here we assume perceived speed $\hat{s}$ to be the mode of the posterior. We consider a model with and without response normalization across channels (red dashed line).

function $p(r_i|s)$. Assuming independent channel noise, we can formulate the posterior probability of an Bayesian observer model that performs optimal integration as

$$p(s|\vec{r}) \propto p(s) \prod_i p(r_i|s) \ . \tag{2}$$

We rely on the results of Stocker and Simoncelli [16] for the characterization of the likelihood functions and the speed prior. Likelihoods are assumed to be Gaussians when considered in a transformed logarithmic speed space of the form $s = \log(1 + s_{\text{linear}}/s_0)$, where $s_0$ is a small constant [9]. If we assume that each channel represents a large number of similarly tuned neurons with Poisson firing statistics, then the average channel likelihood is centered on the value of $s$ for which the activity in the channel peaks, and the width of the likelihood $\sigma_i$ is inversely proportional to the square-root of the channel's response [11]. Also based on [16] we *locally* approximate the logarithm of the speed prior as linear, thus $\log(p(s)) = as + b$.

For reasons of simplicity and without loss of generality, we focus on the case where the stimulus activates two channels with responses $\vec{r} = [r_i], i \in \{1, 2\}$. Given our assumptions, the likelihoods are normal distributions with mean $\mu(r_i)$ and standard deviation $\sigma_i \propto 1/\sqrt{r_i}$. The posterior (2) can therefore be written as

$$p(s|\vec{r}) \propto \exp\left[-\frac{(s - \mu(r_1))^2}{2\sigma_1^2} - \frac{(s - \mu(r_2))^2}{2\sigma_2^2} + as + b\right] \ . \tag{3}$$

We assume that the model observer's speed percept $\hat{s}$ reflects the value of $s$ that maximizes the posterior. Thus, maximizing the exponent in Eq. 3 leads to

$$\hat{s} = \frac{\sigma_2^2}{\sigma_1^2 + \sigma_2^2}\mu(r_1) + \frac{\sigma_1^2}{\sigma_1^2 + \sigma_2^2}\mu(r_2) + a\frac{\sigma_1^2\sigma_2^2}{\sigma_1^2 + \sigma_2^2} \ . \tag{4}$$

A full probabilistic account over many trials (observations) requires the characterization of the full distribution of the estimates $p(\hat{s}|s)$. Assuming that $E\langle\mu(r_i)|s\rangle$ approximates the stimulus speed $s$, the expected value of $\hat{s}$ is

$$E \langle \hat{s}|s \rangle = \frac{\sigma_2^2}{\sigma_1^2 + \sigma_2^2} E \langle \mu(r_1)|s \rangle + \frac{\sigma_1^2}{\sigma_1^2 + \sigma_2^2} E \langle \mu(r_2)|s \rangle + a \frac{\sigma_1^2 \sigma_2^2}{\sigma_1^2 + \sigma_2^2}$$

$$= \frac{\sigma_2^2}{\sigma_1^2 + \sigma_2^2} s + \frac{\sigma_1^2}{\sigma_1^2 + \sigma_2^2} s + a \frac{\sigma_1^2 \sigma_2^2}{\sigma_1^2 + \sigma_2^2} = s + a \frac{\sigma_1^2 \sigma_2^2}{\sigma_1^2 + \sigma_2^2} \ . \tag{5}$$

Following the approximation in [16], the variance of the estimates $\hat{s}$ is

$$var \langle \hat{s}|s \rangle \approx \left( \frac{\sigma_2^2}{\sigma_1^2 + \sigma_2^2} \right)^2 var \langle \mu(r_1)|s \rangle + \left( \frac{\sigma_1^2}{\sigma_1^2 + \sigma_2^2} \right)^2 var \langle \mu(r_2)|s \rangle$$

$$= \left( \frac{\sigma_2^2}{\sigma_1^2 + \sigma_2^2} \right)^2 \sigma_1^2 + \left( \frac{\sigma_1^2}{\sigma_1^2 + \sigma_2^2} \right)^2 \sigma_2^2 = \frac{\sigma_1^2 \sigma_2^2}{\sigma_1^2 + \sigma_2^2} \ . \tag{6}$$

The noisy observer's percept is fully determined by Eqs. (5) and (6). By a similar derivation it is also easy to show that for a single active channel the distribution has mean $E \langle \hat{s}|s \rangle = s + a\sigma_1^2$ and variance $var \langle \hat{s}|s \rangle = \sigma_1^2$.

The model makes the following predictions: First, the variance of the speed estimates (*i.e.*, percepts) for stimuli that activate both channels is always smaller than the variances of estimates that are based on each of the channel responses alone ($\sigma_1^2$ and $\sigma_2^2$). This improved reliability is a hallmark of optimal cue combination as has been demonstrated for cross-modal integration [4, 7]. Second, because of the slow speed prior $a$ is negative, and perceived speeds are more biased toward slower speeds the larger the sensory uncertainty. As a result, the perceived speed of combined stimuli that activate both channels is always faster than the percepts based on each of the individual channel responses alone. Finally, the model predicts that the perceived speed of a combined stimulus solely depends on the responses of the channels to its constituent components, and is therefore independent of the relative phase of the components we combined [5].

## 2.1 Response normalization

So far we assumed that the channels do not interact, *i.e.*, their responses are independent of the number of active channels and the overall activity in the system. Here we extend our proposal with the additional hypothesis that channels interact via divisive normalization. Divisive normalization [6] has been considered one of the canonical neural computations responsible for *e.g.*, contrast gain control, efficient coding, attention or surround suppression [13] (see [3] for a comprehensive review). Here we assume that the response of an individual channel $r_i$ is normalized such that its normalized response $r_i^*$ is given by

$$r_i^* = r_i \frac{r_i^n}{\sum_j r_j^n} \ . \tag{7}$$

Normalization typically increases the contrast (*i.e.*, the relative difference) between the individual channel responses for increasing values of the exponent $n$. For large $n$ it typically acts like a winner-takes-all mechanism. Note that normalization affects only the responses $r_i$, thus modulating the width of the individual likelihood functions. The integration based on the normalized responses $r_i^*$ remains optimal (see Fig. 2). By explicitly modeling the encoding of visual motion in spatiotemporal frequency channels, we already extended the Bayesian model of speed perception toward a more physiological interpretation. Response normalization is one more step in this direction.

## 3 Results

In the second part of this paper we test the validity of our model with and without channel normalization against data from a psychophysical two alternative forced choice (2AFC) speed discrimination experiment.

## 3.1 Speed discrimination experiment

Seven subjects performed a 2AFC visual speed discrimination task. In each trial, subjects were presented for 1250ms with a reference and a test stimulus on either side of a fixation mark (eccentricity

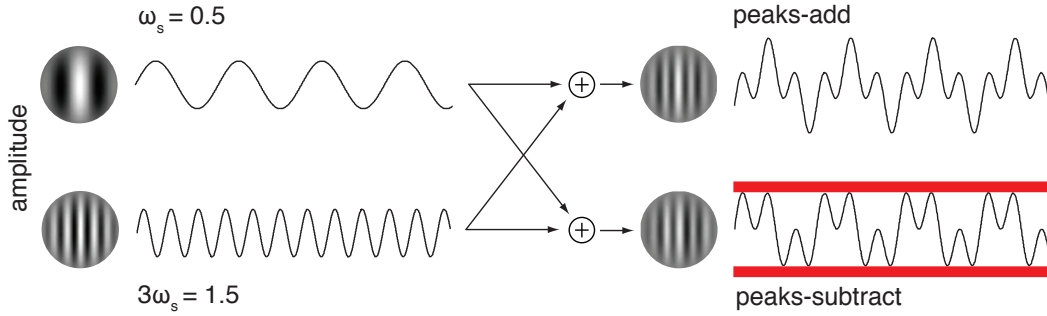

Figure 3: Single frequency gratings were combined in either a "peaks-add" or a "peaks-subtract" phase configuration ($0\,\mathrm{deg}$ and $60\,\mathrm{deg}$ phase, respectively) [5]. The red bar indicates that the two configurations have different overall contrast levels even though they are composed of the same frequencies. We used these two phase-combinations to test whether the channel hypothesis is valid or not.

$6\,\mathrm{deg}$, size $4\,\mathrm{deg}$). Both stimuli were drifting gratings, both drifting either leftwards or rightwards at different speeds. Motion directions and the order of the gratings were randomly selected for each trial. After stimulus presentation, a brief flash appeared on the left or right side of the fixation mark and subjects had to answer whether the grating that was presented on the indicated side was moving faster or slower than the grating on the other side. This procedure was chosen in order to prevent potential decision biases.

The stimulus test set comprised 10 stimuli. Four of these stimuli were simple sinewave gratings of a single spatial frequency, either $\omega_s = 0.5$ or $3\omega_s = 1.5\,\mathrm{c/deg}$. The low frequency test stimulus had a contrast of $22.5\%$, while the three higher frequency stimuli had contrasts 7.5, 22.5 and $67.5\%$, respectively. The other six stimuli were pair-wise combinations of the single frequency gratings (Fig. 3), combined in either a "peaks-add" or a "peaks-subtract" phase configuration [5] (*i.e.* $0\,\mathrm{deg}$ and $60\,\mathrm{deg}$ phase). All test stimuli were drifting at a speed of $2\,\mathrm{deg/s}$. The reference stimulus was a broadband stimulus stimulus whose speed was regulated by an adaptive staircase procedure. Each of the 10 stimulus conditions were run for 190 trials. Data from all seven subjects were combined.

The simple stimuli were designed to target individual spatiotemporal frequency channels while the combined stimuli were meant to target two channels simultaneously. The two phase configurations (peaks-add and peaks-subtract) were used to test the multiple channel hypothesis: if combined stimuli are decomposed and processed in separate channels, their perceived speeds should be independent of the phase configuration. In particular, the difference in overall contrast of the two configurations should not affect perceived speed (Fig 3).

Matching speeds (PSEs) and relative discrimination thresholds (Weber-fraction) were extracted from a maximum-likelihood fit of each of the 10 psychometric functions with a cumulative Gaussian. Fig. 4a,b shows the extracted discrimination thresholds and the relative matching speed, respectively. The data faithfully reproduce the general prediction of the Bayesian model for speed perception [16] that perceived speed decreases with increasing uncertainty, which can be nicely seen from the inverse relationship between matching speeds and discrimination thresholds for each of the different test stimuli. We found no significant difference in perceived speeds and thresholds between the combined grating stimuli in "peaks-add" and "peaks-subtract" configuration (Fig. 4a,b; right), despite the fact that the effective contrast of both configurations differs significantly (by 30, 22 and 11% for the {22.5, 7.5}, {22.5, 22.5} and {22.5, 67.5}% contrast conditions, respectively). This suggests that the perceived speed of combined stimuli is independent of the relative phase between the individual stimulus components, and therefore is processed in independent channels.

## 3.2 Model fits

In order to fit the model observer to the data, we assumed that on every trial of the 2AFC task, the observer first makes individual estimates of the test and the reference speeds $[\hat{s}_t, \hat{s}_r]$ according to the corresponding distributions $p(\hat{s}|s)$ (see Section 2), and then, based on these estimates, decides

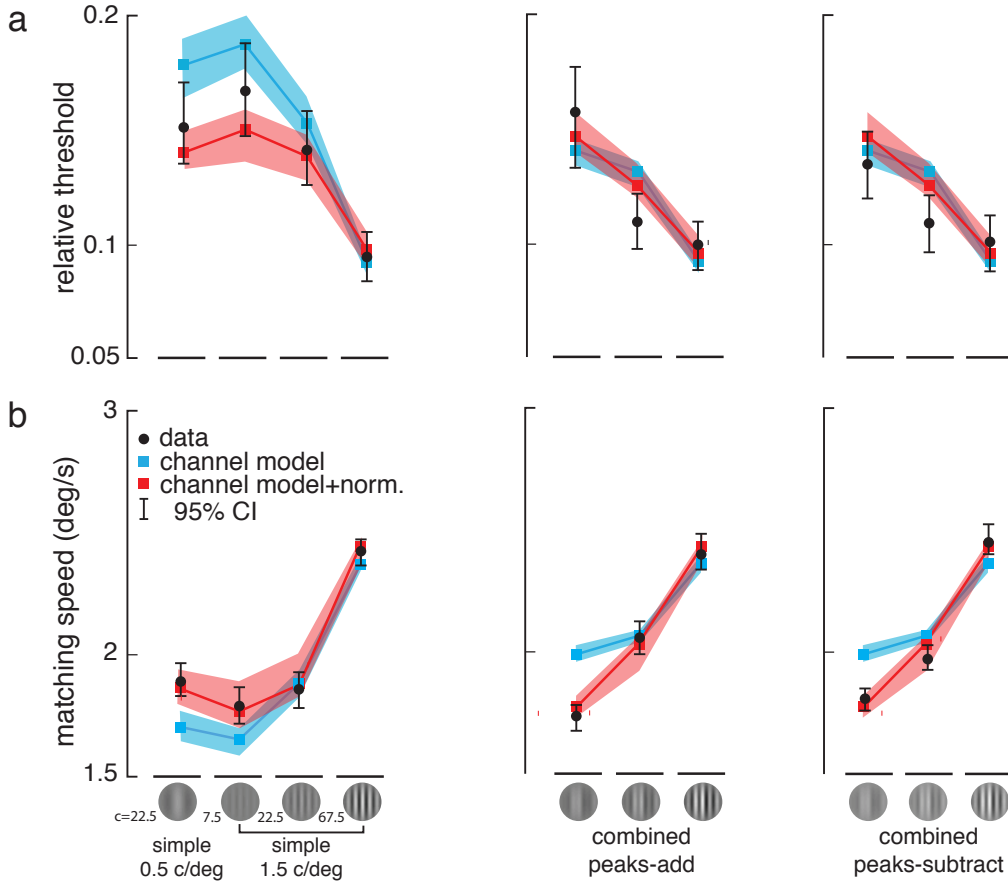

Figure 4: Data and model fits for speed discrimination task: a) relative discrimination thresholds (Weber-fraction) and b) matching speeds (PSEs). Error bars represent the 95% confidence interval from 100 bootstrapped samples of the data. For the single frequency gratings, the perceived speed increases with contrast as predicted by the standard Bayesian model. For the combined stimuli, there is no significant difference (based on 95% confidence intervals) in perceived speeds between the combined grating stimuli in "peaks-add" and "peaks-subtract" configuration. The Bayesian model with normalized responses (red line) better accounts for the data than the model without interaction between the channels (blue line).

which stimulus is faster. According to signal detection theory, the resulting psychometric function is described by the cumulative probability distribution

$$P(\hat{s}_r > \hat{s}_t) = \int_0^\infty p(\hat{s}_r|s_r) \int_0^{\hat{s}_r} p(\hat{s}_t|s_t)d\hat{s}_t d\hat{s}_r \tag{8}$$

where $p(\hat{s}_r|s_r)$ and $p(\hat{s}_t|s_t)$ are the distributions of speed estimates for the reference and the test stimulus according to our Bayesian observer model. The model without normalization has six parameters: four channel responses $r_i$ for each simple stimulus reflecting the individual likelihood widths, the reference response $r_{\text{ref}}$ and the local slope of the prior $a$.[1] The model with normalization has two additional parameters $n_1$ and $n_2$, reflecting the exponents of the normalization in each of the two channels (Eq. 7).

The model with and without response normalization was simultaneously fit to the psychometric functions of all 10 test conditions using the cumulative probability distribution (Eq. 8) and a

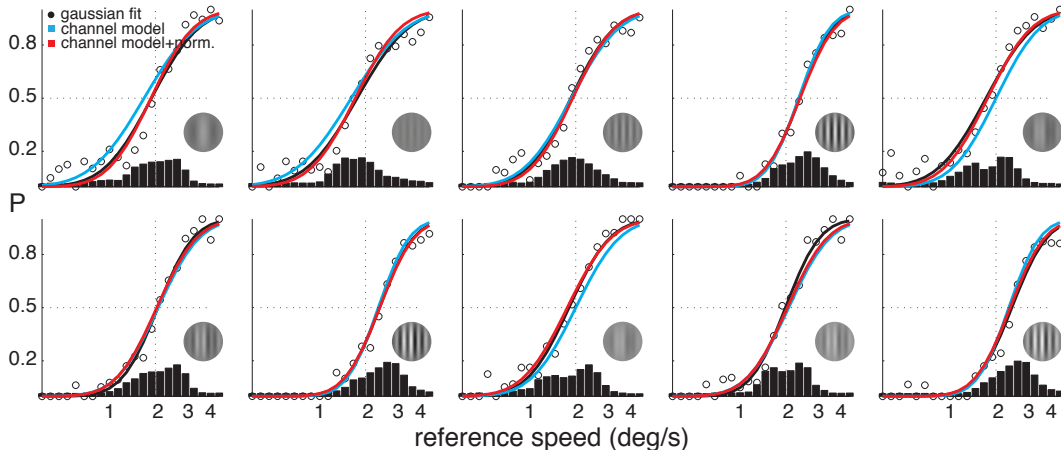

Figure 5: Psychometric curves for the ten testing conditions in Figure 4 (upper left to lower right corner): Gaussian fits (black curves) to the psychometric data (circles) are compared to the fits of the Bayesian channel model (blue curves) and the Bayesian channel model with normalized responses (red curves). Histograms reflect the distributions of trials for the average subject.

maximum-likelihood optimization procedure. Figure 5 shows the fitted psychometric functions for both models as well as a generic cumulative Gaussian fit to the data. From these fits we extracted the matching speeds (PSEs) and relative discrimination thresholds (Weber-fractions) shown in Fig. 4. In general, the Bayesian model is quite well supported by the data. In particular, the data reflect the inverse relationship between relative matching speeds and discrimination thresholds predicted by the slow-speed prior of the model. The model with response normalization, however, better captures subjects' precepts in particular in conditions where very low contrast stimuli were combined. This is evident from a visual comparison of the full psychometric functions (Fig. 5) as well as the extracted discrimination thresholds and matching speeds (Fig. 4). This impression is supported by a log-likelihood ratio in favor of the model with normalized responses. Computing the Akaike Information Criterion (AIC) furthermore reveals that this advantage is not due to the larger number of free parameters of the normalization model with an advantage of $\Delta$AIC = 127 (with significance $p = 10e - 28$) in favor of the latter. Further support of the normalized model comes form the fitted parameter values: for the model with no normalization, the response level of the highest contrast stimulus $r_4$ was not well constrained[2] ($r_1$=6.18, $r_2$=5.50, $r_3$=8.69, $r_4$= 6e+07, $r_{\text{ref}}$=11.66, $a$=-1.83), while the fit to the normalized model led to more reasonable parameter values ($r_1$=10.33, $r_2$=9.96, $r_3$=11.99, $r_4$=37.73, $r_{\text{ref}}$=13.44, $n_1$=2e-16, $n_2$=6.8, $a$=-3.39). In particular, the fit prior slope parameter is in good agreement with values from a previous study [16]. Note that the exponent $n_1$ is not well-constrained because the stimulus set only included one contrast level for the low-frequency channel.

The results suggest that the perceived speed of a combined stimulus can be accurately described as an optimal combination of sensory information provided by individual spatiotemporal frequency channels that interact via response normalization.

## 4  Discussion

We have shown that human visual speed perception can be accurately described by a Bayesian observer model that optimally combines sensory information from independent channels, each sensitive to motion energies in a specific spatiotemporal frequency band. Our model expands the previously proposed Bayesian model of speed perception [16]. It no longer assumes a single likelihood function affected by stimulus contrast but rather considers the combination of likelihood functions based on the motion energies in different spatiotemporal frequency channels. This allows the model to account for stimuli with more complex spatial structures.

We tested our model against data from a 2AFC speed discrimination experiment. Stimuli consisted of drifting sinewave gratings at different spatial frequencies and combinations thereof. Subjects' perceived speeds of the combined stimuli were independent of the phase configuration of the constituent sinewave gratings even though different phases resulted in different overall contrast values. This supports the hypothesis that perceived speed is processed across multiple spatiotemporal frequency channels (Graham and Nachmias used a similar approach to demonstrate the existence of individual spatial frequency channels [5]). The proposed observer model provided a good fit to the data, but the fit was improved when the channel responses were assumed to be subject to normalization by the overall channel response. Considering that divisive normalization is arguably an ubiquitous process in neural representations, we see this result as a consequence of our attempt to formulate Bayesian observer models at a level that is closer to a physiological description. Note that we consider the integration of the sensory information still optimal albeit based on the normalized responses $r_i^*$. Future experiments that will test more stimulus combinations will help to further improve the characterization of the channel responses and interactions.

Although we did not discuss alternative models, it is apparent that the presented data eliminates some obvious candidates. For example, both a *winner-take-all model* that only uses the sensory information from the most reliable channel, or an *averaging model* that equally weighs each channel's response independent of its reliability, would make predictions that significantly diverge from the data. Both models would not predict a decrease in sensory uncertainty for the combined stimuli, which is a key feature of optimal cue-combination. This decrease is nicely reflected in the measured decrease in discrimination thresholds for the combined stimuli when the thresholds for both individual gratings were approximately the same (Fig. 4b). Note, that because of the slow speed prior, a Bayesian model predicts that the perceived speed are inversely proportional to the discrimination threshold, a prediction that is well supported by our data. The fitted model parameters are also in agreement with previous accounts of the estimated shape of the speed prior: the slope of the linear approximation of the log-prior probability density is negative and comparable to previously reported values [16].

In this paper we focused on speed perception. However, there is substantial evidence that the visual system in general decomposes complex stimuli into their simpler constituents. The problem of how the scattered information is then integrated into a coherent percept poses many interesting questions with regard to the optimality of this integration across modalities [4, 7]. Our study generalizes cue-integration to the pooling of information within a single perceptual modality. Here we provide a behavioral account for both discrimination thresholds and matching speeds by directly estimating the parameters of the likelihoods and the speed prior from psychophysical data.

Finally, the fact that the Bayesian model can account for both the perception of simple and complex stimuli speaks for its generality. In the long term, the goal is to be able to predict the perceived motion for an arbitrarily complex natural stimulus, and we believe the proposed model is a step in this direction.

### Acknowledgments

This work was supported by the Office of Naval Research (grant N000141110744).

## Footnotes

[1] Alternatively, channel responses as function of contrast could be modeled according to a contrast response function $r_i = M + R_{\text{max}} \frac{c^2}{c_{50}^2 + c^2}$, where M is the baseline response, $R_{\text{max}}$ the maximal response, and $c_{50}$ is the semi saturation contrast level.

[2]The fit essentially assumed $\sigma_4 = 0$.

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
