[Reviews · NeurIPS 2013]

Submitted by Assigned_Reviewer_4

COMMENTS BASED ON REVIEWER DISCUSSIONS AND AUTHOR REBUTTAL:

I agree with the other reviewers that more could be done to constrain the specifics of the cue integration mechanism. However, I believe that if the data set is expanded, allowing the models to be better constrained, then the paper is appropriate and interesting for the NIPS community.

I have left my quality score as it was, but I agree with the other reviewers that the paper merits a ``1'' rather than a ``2'' for impact score.

ORIGINAL REVIEW:
Summary:
This paper extends an existing model for the perception of visual speed that uses a Bayesian observer model acting on the activity of independent spatiotemporal frequency channels. Previously, the model accounted for illusions of perceived speed by postulating the Bayes-optimal combination of noisy sensory representations with a prior for slow speeds. Here this model is extended to operate on spatiotemporal filter outputs, allowing the model to account for changes in perceived speed as a function of spatial frequency. The revised model is shown to provide a good fit to human speed discrimination data.

Strengths:
The model is elegantly simple and provides a good fit to data. The model is a useful extension of previous work, providing a more general and neurophysiologically plausible version. The behavioral experiment is straightforward and clearly makes the case for the addition of the normalization mechanism. The paper is concise and clearly written.


Weaknesses:

-- ``The perception of combined stimuli can therefore be fully predicted by optimal combining responses from individual spatiotemporal frequency channels.''
This statement (``fully predicted'') seems too strong to me for two reasons: 1) while bias is predicted really well, threshold fits are less impressive and 2) no new data has been predicted (all data were used in the model fits). It would be interesting to see the cross-validated area under the ROC between the two models and the Gaussian fits to give the reader a better sense of the scale of the prediction improvement.

On thresholds, while it's hard to say from three data points, is there a plausible reason for my impression that the discrimination thresholds for combined stimuli seem to decline more steeply before plateauing (e.g. power law or exponential) rather than the more linear predictions of the model? That is, the threshold predictions at high contrasts are good, but thresholds at low contrasts might be under-predicted whereas thresholds at medium contrasts are over-predicted. Would the ``Foley function'' with the exponent of the numerator set to 2.4 (instead of 2 in your CRF) provide a better fit to those data due to the accelerating contrast response early in the function?

If I understand correctly, the channel variances \sigma^2 correspond to (or can be transformed to give) the bandwidths of the hypothetical channels. Can the authors comment on whether the bandwidths estimated in their model fits correspond to bandwidths of spatiotemporally-oriented filters measured elsewhere (e.g. physiology, psychophysical masking paradigms)?

Line 367: ``The model also fully supports previous accounts on the shape of the prior for speed: the slope of the linear approximation of the prior probability density is negative and conforms to previously reported values.'' Where in the manuscript is the slope of the prior reported (or any fitted model parameters for that matter)? A small table of best-fitting model parameters and their standard errors would be of use here.

-- How easily can the two-channel model derivation be extended? Will multi-channel (> 3) responses require numerical approximations?

Minor points:

Line 85: grammar... "proposing that the visual system treats the responses in individual spatiotemporal frequency channel as independent cues".
Line 103: grammar... "only those unit" should be "units"
Line 107: grammar... "An moving stimulus" should be "A moving stimulus"
Line 131: typo "r2,"
Line 242: typo "agains data"

I found the paragraph beginning Line 256 unclear, and I didn't really understand the stimulus combinations until reading the results. I think it would be clearer to write something like:
``There were four simple stimuli: a 0.5 c/deg grating presented at 22.5% contrast, and three 1.5 c/deg gratings presented at 7.5, 22.5 and 67.5% contrast respectively. Combined stimuli were superpositions of the lower and higher frequency simple stimuli, in either a ``peaks-add'' or ``peaks-subtract'' configuration.''

Line 261:``We used a broadband stimulus regulated by an adaptive staircase as reference.'': it would be useful to explicitly state here that it is the speed of the reference that is under adaptive control.

Line 315: ``we plot the peaks-add conditions as black squares, and the peaks-subtract conditions as black triangles.'' I see no black triangles in the figure; I assume these were separated into separate panels.

Line 320: ``independent on phase'' should be ``independent of phase''

Line 375: ``This decrease is nicely reflected in the measured discrimination thresholds for the combined stimuli where the threshold for both individual gratings were of the same size (Fig. 4b)'': I think this figure reference must be wrong because 4b refers to bias, not thresholds.

Line 377: ``predicts that the perceived speed are'' should be ``speeds are'' or ``speed is''.

Line 378: ``sensory system in general processes'' should be ``systems in general processes'' or ``system in general process''. Same for ``question'' on line 380.

Summary: A clearly-written paper fitting an elegant model of channel combination in the context of human speed discrimination, combined with solid psychophysics.

Submitted by Assigned_Reviewer_5

The paper takes a step towards understanding visual motion processing of complex, multidimensional stimuli. The approach taken by the authors is to decompose one dimension of the stimulus space (ie spatial frequency) into channels containing independent information about the other stimulus dimension (ie speed), and so solve a problem analogous to multisensory cue integration.


- Quality
First, the authors derive the predictions from a Bayesian ideal observer model that integrates the cues optimally (ie, weights each cue by its reliability). The Bayesian model is sound, extending previously published work (ref 15). The variant with divisive normalization makes sense, but the consequences of the nonlinearity on the probabilistic representation should be explored and described more: Exactly how does the posterior variance depend on contrast after vs before normalization? What are the implications for the predicted biases and thresholds?

Then the authors compare model predictions with psychophysical data. They conclude that information is indeed processed independently, and then combined optimally, across frequency channels. Furthermore, they claim that divisive normalization improves the fits to the data. However, the data shown do not provide strong evidence to support those claims, and the incomplete analysis does not entirely rule out simpler models, as explained below.

The main prediction of optimal cue combination is that the variance of the estimate from the combined cues (superposition of two spatial frequencies) is smaller than for each individual cue alone, and the authors state in the abstract that: "the perceptual biases and discrimination thresholds are always smaller for the combined stimuli, supporting the cue combination hypothesis."
However, 1) the thresholds did not decrease when contrasts were not matched (first and third data points in 4a-center, right, compared to 4a-left; this is also true for the biases in 4b), which accounts for 4 out of 6 data points! 2) None of the fitted models captured the decrease in threshold when it was present in the data (second data point in 4a-center, right)! 3) given 1 and 2, there is also no evidence for the following claim "Although we did not discuss alternative models, it is apparent that the data from our psychophysical experiments eliminates some obvious candidates. For example, a winner-take-all model, where the observer only uses the most reliable cue, or the averaging model, where cues are weighted in equal ratios, would significantly diverge from the data. Both these models do not predict a decrease in uncertainty for the combined stimulus, which is a key feature of optimal cue-combination." I think the authors should show those simpler models.

The claim that normalization improves the fits is also not quantified, and the reason for the supposed improvement is not explained. From Figure 4 alone, it is hard to judge by eye which model does better. It is true that "this model [without normalization] only partially fits the data, with significant discrepancy in test conditions where the difference in contrast of component stimuli is high." But the model with normalization is a worse fit to (and deviates significantly from) the data for the simple stimuli with high SF (fig 4a); and it fails significantly for the combined stimuli with matched contrast.

- Clarity
The paper is clearly written, well organized, and contains all the information necessary to understand the model and the results. Here are some minor suggestions:
line 131, typo: "r2" should be "r_2"
around line 131: state that r_i is deterministic, there is no intrinsic channel noise
line 144: "a ’channel’ represents a large number of similarly tuned neurons" I assume they are similarly tuned wrt frequency, but span the entire range of speeds; this should be stated explicitly to avoid confusion.
line149: "the change in the slope of the prior within a likelihood width centered at speed s is expected to be small" can only be true if s is far from the peak of the prior.
line 150: "the prior by its logarithm" should be "the logarithm of the prior"
line 315: "we plot the peaks-add conditions as black squares, and the peaks-subtract
conditions as black triangles." There are only black circles in the plot.
line 333: "r_ref" not defined.
lins 332-337: which of the two models was actually used?
line 376: "Fig. 4b" should be "Fig. 4a"

- Originality
This is an interesting extension of previously published work (ref 15) that uses simple analytical results derived in refs 4,10.

- Significance
The paper addresses an important problem, that of visual motion processing of complex, multidimensional stimuli, and takes a step in that direction by proposing to decompose complex stimuli in independent channels, and then apply the results of optimal cue combination. It will be interesting to see how the approach can be scaled to more complex and realistic stimuli.

Summary: The paper proposes to treat visual motion processing of complex stimuli as optimal cue combination across independent frequency channels. The approach is an interesting extension of previous work, but both the experimental results and data analysis presented by the authors provide only weak support.

Submitted by Assigned_Reviewer_6

The paper demonstrates that human speed perception can be described by a Bayesian model that 1) optimally combines signals from normalized spatiotemporal channels, and 2) incorporates a prior that prefers slow speed. It also shows that the model with normalization fits human performance better than the model without normalization does. The interesting part of the psychophysical results is that speed perception does not ALWAYS depend on contrast (lines 316-319). When the effective contrast of a composite grating is modulated by the phase difference between the underlying gratings, perception of speed does not seem to be different. This trend is nicely captured by their model. This finding seems to suggest that the contrast-dependent phenomenon of speed perception originates from individual spatiotemporal channels’ contrast-dependent responses, but not from a later stage of processing that integrates the signals.
The model is straightforward and well-presented. The model with divisive normalization makes a good bridge to connect to physiology. But I don’t understand why the authors used different exponents for different channels in the normalization model (line 338). This seems to be different from what standard divisive normalization models do. Any advantage of using different exponents? Would the two parameters play an important role in providing good fittings? I also concerned about the results of model comparisons. It is not clear whether the model with normalization (red) or the one without (blue) is really“better”. Although the red model fits perceived speed better, the blue model appears to do better in fitting thresholds. The authors should report some objective measures of goodness of fit (e.g., AIC or BIC) to compare the two models before jumping into conclusion that “… the normalized model … better captures human behavior… (line 345)”.
I also have one comment on psychophysics. The author found that the thresholds were lower for the combined gratings than for the simple gratings. However, this comparison may not be fair, because a combined grating (constructed by superposition, line 259) has a higher effective contrast than its underlying simple gratings, regardless of whether it is peaks-add or peaks-subtract. For a more fair comparison, the authors should use simple gratings with a contrast level that is equal to the effective contrast the combined gratings (or at least equal to the lower one between the two combined gratings). Without these findings, it is hard to conclude whether the decrease in threshold is due to a benefit from cue combination, or simply due to higher effective contrast in the combined gratings. This would undermine the argument to rule out other models, as discussed in the third paragraph in Discussion.
Minor points:
I recommend the authors to report the fitted parameters, so that 1) readers can have a better understanding about what each parameter is doing. E.g., The magnitude of the local slope of the prior a can roughly imply how important “slowness” is in the model; and 2) future researchers can follow up with the model based on the explicitly reported parameter values.
The data plotted in Figure 4 represent the “average observer” (line 269). While the absolute values of thresholds and biases may differ across individuals, the authors should at least report whether a trend that is similar to that observed in the “average observer” was consistently observed across individual observers.
What is meant by “relative speed biases”? At what percentage correct was the threshold determined?
Lines 376: I suppose it’s Figure 4a instead of 4b, as it’s about thresholds, not speed bias
Lines 315-316: It seems that the “black squares/triangles” described in this sentence are not present in the figures.
Summary: This is a solid paper to combine both computational model and psychophysical experiments to understand human motion perception. Although the work is a bit incremental, I believe this work is of sufficient theoretical merit.
Author Feedback

Author rebuttal: We would like to thank the reviewers for their effort in reviewing our submission and for their valuable and constructive feedback. Below listed are our replies to each reviewer's main concerns:

Assigned_reviewer_4 comments that our statement that the model “fully predicts” the data is too strong because i) we do not predict new data based on a fit model, and ii) the fits are not perfect. We did use the term "prediction" rather deliberately and agree that this might be misleading; we will change that in the revised version of the paper (e.g. "accounts for"). Regarding the quality of the fit, we would like to point out that our model accounts for the full psychometric curves of the performed 2AFC experiments and is fit using maximum-likelihood procedure. The plotted bias and discrimination thresholds therefore only represent summary characteristics of the behavior, and while the thresholds are much less constrained by the maximum-likelihood fit than the bias the qualitative fits of both models are well accounting for the data. We think that the quantitative imperfections of the fit are due to assumptions we make (e.g. Gaussian noise) that are all reasonable, but not necessarily true. We fully agree that a formal comparison of the two models and a table with fitted parameters provides useful information which we will include in the revised paper. Finally, we are currently extending our dataset with more contrast levels that allow a better quantitative characterization of the channels
and the normalization parameters. It is our ultimate goal to acquire behavioral data for more complex stimuli and test models with more than two channels.


Assigned_reviewer_5 observes that only two out of six thresholds for the combined stimuli show a substantial decrease compared to the thresholds of the individual component stimuli. We would like to point out that such behavior is typical for optimal (Bayesian) integration: the largest decrease in threshold is expected when individual cues have the same level of uncertainty (i.e. threshold) whereas the decrease in threshold can be small or negligible if the difference in cue uncertainty is large. This is well reflected in the presented data. We acknowledge, however, that the current dataset is limited in the sense that it only provides two conditions of integration where a large improvement in threshold is expected. As mentioned above, we are currently expanding the dataset and aim to include more conditions where optimal integration predicts larger decreases in threshold. We note that our
proposed model well predicts the inverse relationship between bias and threshold (due to the slow speed prior), which is fairly strong evidence in favor of a Bayesian model of cue integration. We agree, however, that the current dataset does not provide strong constraints of the exact form of integration. The current expansion of the dataset will allow us to further distinguish between different models in this context.

Assigned_reviewer_6 first comments on the need for two exponents in the normalization model rather than having one exponent for both channels. Unfortunately and as already mentioned in the responses to the reviewers above, our current database can not fully constrain all aspects of the normalization model, particularly given that data for one of the channels is limited to one contrast level. We expect that the new dataset we are currently collecting will allow us to better constrain the single and combined responses resulting in a common exponent for both channels. The reviewer also comments on the lack of a quantitative model comparison. As mentioned above, we will include such comparison in the revised version of the paper. Last, the reviewer argues that the combined stimuli in our experiment have higher contrast and therefore a decrease in threshold is expected even in the absence of cue combination. We show that the perceived speed of the combined stimuli does not depend on the total contrast by demonstrating the independence on phase between the two simple cues. We agree, however, that testing complex stimuli with the same total contrast as simple stimuli would be a
striking demonstration, and we will consider this option for future work. As previously suggested, we will add the values of the fit model parameters and a quantitative model comparison in the revised paper.

Finally, we thank all three reviewers for the careful reading and their numerous minor comments that we will incorporate in the revision of the paper.

In summary, we think that our work proposes an interesting extension of the Bayesian model of speed perception. We demonstrate that the perceived speed of more complex stimuli is well described by a Bayesian observer that combines information form different spatiotemporal frequency channels. We agree, however, that the current dataset might not provide sufficiently strong constraints to absolutely determine the specifics of the integration mechanism. We are currently expanding our dataset to resolve this issue.